# Multidimensional characteristics of young Brazilian volleyball players: A Bayesian multilevel analysis

**Felipe G. Mendes**[1], **Ahlan B. Lima**[1], **Marina Christofoletti**[1], **Ricardo T. Quinaud**[1], **Carine Collet**[2], **Carlos E. Gonçalves**[3], **Humberto M. Carvalho**[1]*

**1** Department of Physical Education, School of Sports, Federal University of Santa Catarina, Florianópolis/Santa Catarina, Brazil, **2** College of Health Sciences and Sport Science, Santa Catarina State University, Florianópolis/Santa Catarina, Brazil, **3** Faculty of Sport Sciences and Physical Education, University of Coimbra, Coimbra, Portugal

* hmoreiracarvalho@gmail.com

## Abstract

Brazil has been the benchmark for volleyball performance for at least two decades, providing a unique context to examine expertise development. This study examined the variation in body size, functional capacities, motivation for achievement, competitiveness, and deliberate practice of youth volleyball players associated with differences in biological maturity status, chronological age, and accumulated deliberate volleyball practice, adopting a Bayesian multilevel modeling approach. We considered 68 female and 94 male adolescent players (14.2 years, 90% confidence interval: 12.7 to 16.0). Players were grouped by the onset of deliberate volleyball practice as related to biologic maturation milestones [pre-puberty deliberate practice onset (12% of the sample), mid-puberty deliberate practice onset (51% of the sample), and late-puberty deliberate practice onset (37% of the sample). There was substantial variation in body dimensions and functional performance by gender. There was no variation by gender for motivation for deliberate practice and motivation for achievement and competitiveness. The young volleyball players appeared to be highly motivated and committed to deliberate practice, achievement, and competitiveness. Alignment of chronological age, biological maturation, and accumulated training experience allow more in-depth insights into young volleyball players' development, providing sounder support for coaches´ decisions.

## Introduction

The development of young volleyball players into adult sport expertise is likely nonlinear and dependent on many interacting factors to attain optimal physical, technical, tactical, and behavioral characteristics [1]. Hence, the use of multidimensional approaches to understanding young athletes' development is recommended [2–4]. A key issue in youth sports lies in interpreting athletes´ performance, which is generally aligned by chronological age. Interpretations based on chronological age *per se* are incomplete, at best. Often, coaches, researchers,

**Data Availability Statement:** The datasets, model codes and supplementary material supporting the

conclusions of this article are available in the OSF repository (osf.io/ud2ev).

**Funding:** FGM, ABL, MC and RTQ were supported by grants from the Coordenação de Aperfeiçoamento de Pessoal de Nível Superior – Brasil (CAPES) – Finance Code 001. The funders had no role in study design, data collection and analysis, decision to publish, or preparation of the manuscript.

**Competing interests:** The authors have declared that no competing interests exist.

and interested stakeholders infer about players' growth and maturity status based on chronological age. However, there is wide variability in the development of body size, functions, and behavior during pubertal growth [4, 5]. On the other hand, youth sports programs often assume that sport expertise is positively related to the accumulated number of hours of practice [6, 7]. The variability of accumulated deliberate practice, i.e., accumulated experience in the sport, between players should be considered when interpreting their development.

Movement patterns in volleyball require high-intensity efforts with an intermittent nature, i.e., frequent short bouts of high-intensity exercise followed by periods of low-intensity activity and brief rest periods [8]. The match duration is about 90 minutes. Hence, it requires from players a good fitness level to sustain efforts requiring mainly aerobic and anaerobic alactic energy systems [9–11]. Considerable demands are also placed on the neuromuscular system during the various sprints and jumps (blocking and spiking) and high-intensity court movement that repeatedly occurs during the match [12]. Overall, volleyball players are expected to express high levels of speed, agility, upper-body and lower-body muscular power [12, 13]. However, relatively little is known about volleyball players' physical and functional characteristics, particularly during adolescence.

Nowadays, the early onset of deliberate practice has become a mainstream path for talent development. Young athletes are becoming engaged and committed to youth sports programs at early pre-pubertal ages [14, 15]. The engagement in youth sports development programs is probably conditional on a strong orientation towards competitive success, added to a strong will to attain expertise and commitment to high practice volume and intensity [7, 16]. Nevertheless, it remains unclear whether the early onset of deliberate practice positively or negatively affects young athletes' psychosocial characteristics, particularly achievement and competitiveness motivation and motivation for deliberate practice [17–20].

Interpretations of young athletes' performance or behavior are dependent on individual (e.g., gender, accumulated deliberate practice in the sport, or maturity status) and contextual characteristics (e.g., age-group category or competitive level) [21]. Hence, youth sports observations need to consider cross-classified nesting within and between groups, which often requires coping with an imbalance in sample size and heterogeneity among players. Traditional single-level regression models have been used to deal with the data, especially in settings with a low group-level variation where multiple comparisons are a particular concern [22]. Multilevel models can cope with imbalanced samples and explicitly assume clusters of observations within the data with unique coefficients [23]. The estimates for each cluster take advantage of the full sample information (i.e., shifting estimates toward each other), yielding better estimates [22].

Multilevel models can be fitted within a Bayesian framework [24]. Bayesian methods treat parameters as random variables combining both sample data and prior distribution information to estimate a (posterior) probability distribution that reflects the uncertainty associated with how well they are known, based on the data [25, 26]. Those unfamiliar with Bayesian methods will not see significance tests, but instead, the relative confidence in different models and parameter values is assessed by means, confidence intervals (also referred to as credible intervals), and visual inspection of model predictions [23]. Hence, Bayesian methods allow a direct probabilistic interpretation of confidence intervals and posterior probabilities, relevant in Sport Sciences, where interests often lie in estimating small effects [25].

The Brazil volleyball national teams have been consistently the highest-ranked nation globally at the male adult level and ranging from the highest to the fourth-ranked country at the female level [27]. Hence, youth volleyball programs in Brazil offer a unique context to study expertise development in team sports. This study examined the variation in body size, functional capacities, motivation for achievement, competitiveness, and deliberate practice of

youth volleyball players associated with differences in biological maturity status, chronological age, and accumulated deliberate volleyball practice, adopting a Bayesian multilevel modeling approach.

## Materials and methods

### Study design and participants

The present survey included 162 young volleyball players (female players, n = 68; male players, n = 94). The players were engaged in formal training and competition within under 13 (n = 35), under 15 (n = 71), and under 17 (n = 56) teams from two clubs from Florianópolis, Santa Catarina, and Curitiba, Paraná. The players competed at the state level supervised by the *Federação Catarinense de Voleibol* and the *Federação Paranaense de Voleibol*, respectively. The competitive season in Brazil typically runs between February/March until November/December. At the time of the study, all the volleyball players regularly trained (~300–400 min/wk) over a 10-month season (February to November). The Research Ethics Committee approved the study of the Federal University of Santa Catarina. Players and their parents or legal guardians provided written informed consent after giving information about the study's nature. Participation was voluntary, and the players could withdraw from the study at any time.

### Procedures

We calculated chronological age to the nearest 0.1 years by subtracting a birth date from the date of testing. The players were grouped into two-year age categories: under 13 (11.0–12.9), under 15 (13.0–14.9) and under 17 (15.0–16.9) years. Deliberate volleyball practice onset was considered the self-reported age when players started formal training and competition, supervised by a coach within a youth volleyball program registered in the state federation, with no participation in practice and competition in other organized sports.

We used the gender-specific maturity offset protocol [28] to determine payers' maturity status. The offset equations estimate time before or after peak height velocity (PHV) based on chronological age and stature. We subtracted the offset estimate from chronological age to estimate each player's age at PHV. Players' estimated age at PHV was contrasted against a gender-specific reference age at PHV. We derived the references for gender-specific age at PHV based on a meta-analysis of longitudinal growth studies summarized elsewhere [29]. The reference age at PHV was 11.9 (90% confidence interval: 11.8,12.0) years and 13.9 (90% confidence interval; 13.8, 14.0) years for girls and boys, respectively [20]. Then we classified players as follows: early maturers (n = 68), when estimated age at PHV was lower than the gender-specific reference age at PHV by more than six months; average maturers (n = 81) when players' estimated age at PHV was within plus/minus six months of the gender-specific age at PHV; late maturers (n = 13), when estimated age at PHV was higher than the gender-specific reference age at PHV by more than six months. Nevertheless, we assume the limitations of the maturity offset protocol [4], particularly at the observed age range's extremes where bias may be more likely [30]. Hence, we allow for the possibility that a player may have been assigned to the wrong maturity status category.

The onset of deliberate volleyball was interpreted relative to pubertal growth milestones, the ages of onset of the pubertal growth spurt, and at PHV [20]. We grouped the players by the onset of deliberate volleyball practice as follows: pre-puberty deliberate volleyball practice onset (n = 17), the players who started practice before the reference age of pubertal growth spurt onset (female: 9.4 years, 90% confidence interval: 9.1 to 9.7; male: 11.1 years, 90% confidence interval: 10.8 to 11.5; mid-puberty deliberate volleyball practice onset (n = 83), the players starting practice between the reference ages of pubertal growth spurt onset age and at

PHV; late-puberty deliberate volleyball practice onset (n = 62), the players starting practice after the reference age at PHV.

Stature was measured with a portable stadiometer (Seca model 206, Hanover, MD, USA) to the nearest 0.1 cm. Body mass was measured with a calibrated portable balance (Seca model 770, Hanover, MD, USA) to the nearest 0.1 kg. Wingspan was measured with a standard tape measure to the nearest 0.1 cm, from middle digit to middle digit with both shoulders abducted to 90˚. Intra-observer technical errors of measurement were 0.23 (90% confidence interval: 0.17 to 0.42) cm for stature, 0.11 kg (90% confidence interval: 0.07 to 0.27) for body mass and 0.18 cm (90% confidence interval: 0.12 to 0.38) for wingspan.

Considering volleyball-specific effort demands [9–11], we examined players' functional capacities by measuring 10-m sprint, vertical jump (countermovement jump), and upper-body muscular power (2-kg medicine ball overhead throw). The players' running speed was evaluated with a 10-m sprint effort [31] using two photocells (Microgate Polifemo, Bolzano, Italy). The timing gates were positioned at the starting point and 10 m after. Players were instructed to run as quickly as possible the 10-m distance from a standing start. Sprint was measured to the nearest 0.01 s. Intra-observer technical error of measurement was 0.05 s (90% confidence interval: 0.04 to 0.07). Lower-body muscular power was estimated using the vertical jump with countermovement [32]. Vertical jump with countermovement performance was examined using the Optojump photocell system (Microgate, Bolzano, Italy). Players started from an upright standing position and were instructed to begin the jump with a downward movement, which was immediately followed by a concentric upward movement, resulting in a maximal vertical jump. During jumping, hands were held on the hips during all phases of the jumping. Three trials were given with a 30 s rest period, and the best trial was retained for analysis. Intra-observer technical error of measurement was 1.5 cm (90% confidence interval: 1.1 to 2.4). Upper-body muscular power was determined using a 2-kg medicine ball overhead throw. The players were positioned with their knees on the floor, holding the ball with the hands at chest level. They performed an overhead throw for the starting position vigorously as far straight forward as she/he could while maintaining the knees on the floor. The longest distance of three attempts was retained for analysis. Intra-observer technical error of measurement was 0.28 m (90% confidence interval: 0.21 to 0.43).

We used the Deliberate Practice Motivation Questionnaire [33, 34] and the Work and Family Orientation Questionnaire [35]. The Deliberate Practice Motivation Questionnaire, initially designed for chess, comprises 18 items rated on a 5-point Likert scale (1 = completely disagree to 5 = completely agree), considering two dimensions of deliberate practice: will to compete and will to excel. We used the adapted version for team-sports, which was previously translated and validated to Portuguese [7]. The Work and Family Orientation Questionnaire, composed of 19 items and rated on a 5-point Likert scale (1 = completely disagree to 5 = completely agree), assesses four dimensions of achievement: personal unconcern, work, mastery, and competitiveness. This study only used the last three subscales, consistent with previous studies with youth sports samples [4, 20, 36].

## Data analysis

Our estimations were based on Bayesian multilevel models considering the variation on body dimensions, functional capacities, and motivation characteristics, adjusting for cross-classified nesting by gender, age group, maturity status, and the onset of deliberate practice among young Brazilian volleyball players. We fitted the models in R [37] using "brms" package [38], which calls Stan [39]. We standardized (z-score) all the outcomes for interpretative convenience and computational efficiency. We used varying-intercept models where each player´s

outcome (intercept) was estimated as a function of his/her age group, estimated maturity status, and the onset of deliberate practice. Hence for player $i$, with indexes $a$, $m$, $d$, and $g$ for age group, maturity status, the onset of deliberate practice, and gender, respectively. The group-level effect terms (also refered to as random effects) and data-level term (also refered as level-1 residuals) were drawn from normal distributions with variances to be estimated from the data:

$$y_i = \beta^0 + \alpha_{a[i]}^{age\ group} + \alpha_{m[i]}^{maturity\ status} + \alpha_{d[i]}^{deliberate\ practice} + \alpha_{g[i]}^{gender} + \epsilon_i$$

$$\alpha_{a[i]}^{age\ group} \sim N(0, \sigma_{age\ group}^2), for\ a = 1,\ 2,\ 3.$$

$$\alpha_{m[i]}^{maturity\ status} \sim N(0, \sigma_{maturity\ status}^2),\ for\ m = 1,\ 2,\ 3.$$

$$\alpha_{d[i]}^{deliberate\ practice} \sim N(0, \sigma_{deliberate\ practice}^2),\ for\ d = 1,\ 2,\ 3.$$

$$\alpha_{g[i]}^{gender} \sim N(0, \sigma_{gender}^2),\ for\ g = 1,\ 2.$$

$$\epsilon_i \sim N(0, \sigma_{y_i}^2)$$

Measurement of young athletes' performance and behaviors are often noisy, and effects are likely small. Hence, we used weakly informative priors to regularize our estimates, a normal prior (0,5) for the intercept (population-level parameter, also referred to as fixed effect) and group-level parameters. For the data-level residuals ($\epsilon_i$), we used the "brms" default prior, Student-$t$ (3, 0, 2.5). Given the standardization of the outcomes using a normal (0,1) prior for the parameters, we state that the group-level estimates are unlikely to be greater than one standard deviation of the outcome. We run four chains for 2,000 iterations with a warm-up length of 1,000 iterations in each model. The convergence of Markov chains was inspected with trace plots. We used posterior predictive checks to be confident in our models and estimations [24].

## Results

Characteristics of the youth Brazilian volleyball players for the total sample and grouped by gender are shown in Table 1. All but two maturity offset values were positive for the female players, and seventy-one from ninety-four maturity offset values were positive for male players. In general, most of the players in the present sample were beyond the age at PHV. Only 13 players were classified as late maturers. The other players were about evenly distributed as early, and average matures. About 11% of the sample players had pre-puberty onset of deliberate volleyball practice, while about 51% and 38% of the players had mid-puberty and late-puberty deliberate volleyball practice onset, respectively.

Estimations and uncertainty (90% and 67% confidence intervals) of the outcomes are plotted by age group and contrasting with age group by maturity status and the onset of deliberate practice. We separated the plots by gender. Our models accounted for variation associated with age group, maturity status, the onset of deliberate practice, and gender. Hence, we can interpret the effects of target groups, accounting for the other group-effects. Supplementary figures and model codes are available as supplementary material (https://osf.io/ud2ev/).

For body dimensions, variation by gender was substantial (Table 1). As expected, male players were taller and heavier than female players, independent of age and maturity. We plotted players' body dimensions against the World Health Organization (WHO) growth references for stature [40]. The WHO growth references are available for body mass only until ten years

**Table 1. Posterior estimations and 90% credible intervals of young Brazilian volleyball players by gender.**

|  | All sample (n = 162) | Female (n = 68) | Male (n = 94) |
|---|---|---|---|
| Chronological age, yrs | 14.2 (14.0 to 14.5) | 14.3 (14.1 to 14.5) | 14.2 (14.0 to 14.4) |
| Maturity offset, yrs | 1.58 (1.35 to 1.81) | 2.45 (2.24 to 2.67) | 0.94 (0.76 to 1.11) |
| Years of training experience, yrs | 1.8 (1.6 to 2.1) | 1.9 (1.6 to 2.1) | 1.8 (1.6 to 2.0) |
| Stature, cm | 171.2 (169.4 to 172.8) | 167.9 (166.2 to 169.6) | 173.5 (172.1 to 175.0) |
| Body mass, kg | 63.2 (61.0 to 65.5) | 61.5 (59.2 to 63.7) | 64.5 (62.6 to 66.4) |
| Wingspan, cm | 175.7 (173.9 to 177.6) | 173.0 (171.1 to 174.9) | 177.7 (176.1 to 179.3) |
| *Performance* |  |  |  |
| Countermovement jump, cm | 25.3 (24.1 to 26.4) | 23.5 (22.3 to 24.6) | 26.7 (25.6 to 27.5) |
| 2-kg medicine ball throw, m | 6.4 (6.1 to 6.6) | 6.0 (5.5 to 6.1) | 6.8 (6.5 to 7.0) |
| Sprint 10-m, s | 2.12 (2.09 to 2.15) | 2.15 (2.12 to 2.18) | 2.10 (2.08 to 2.13) |
| *Deliberate practice motivation* |  |  |  |
| Will to excel, 1–5 | 4.06 (3.92 to 4.20) | 3.95 (3.80 to 4.09) | 4.15 (4.04 to 4.27) |
| Will to compete, 1–5 | 4.49 (4.42 to 4.56) | 4.48 (4.40 to 4.54) | 4.50 (4.43 to 4.56) |
| *Achievement and competitiveness motivation* |  |  |  |
| Mastery, 1–5 | 4.30 (4.20 to 4.40) | 4.28 (4.19 to 4.37) | 4.32 (4.24 to 4.40) |
| Work, 1–5 | 4.46 (4.38 to 4.54) | 4.46 (4.38 to 4.53) | 4.45 (4.39 to 4.52) |
| Competitiveness, 1–5 | 3.77 (3.66 to 3.89) | 3.64 (3.51 to 3.76) | 3.87 (3.77 to 3.97) |

of age [40]. Hence we used the US population growth references for body mass [41]. Overall, the young Brazilian volleyball players compared favorably with the reference samples, mostly above the 75th percentile for stature, with a substantial part of the sample above the 90th percentile of the WHO growth references (Fig 1). As for body mass, the young Brazilian volleyball players showed mostly between the 50th and 75th percentiles for the US population growth references (S1 Fig of S1 File).

The multilevel regression models estimated parameters are summarized in Table 2. Older female and male players had better functional performance than younger players (Figs 2–4). After adjusting for age group, maturity status, and the onset of deliberate volleyball practice, estimates varied between female and male players for upper-body muscular power (female estimate = 5.1 m, 90% CI: 1.5 to 8.8; male estimate = 6.8 m, 90% CI: 3.2 to 10.4) and countermovement jump (female estimate = 21.7 cm, 90% CI: 13.3 to 29.0; male estimate = 25.3 cm, 90% CI: 16.8 to 32.6). However, for sprint performance, variation by gender was trivial (female estimate = 2.17 s, 90% CI: 2.07 to 2.29; male estimate = 2.12 cm, 90% CI: 2.03 to 2.24).

There was no substantial variation by gender and onset of deliberate volleyball practice for the dimensions of deliberate volleyball practice motivation (Table 2). The scores for both will to excel and will to compete showed small variation by age group, at best. There was variation between players by maturity status in the motivation for will to excel (standardized estimates: early maturers = 0.16, 90% CI -0.31 to 0.56; average maturers = -0.12, 90% CI: -0.54 to 0.27; late maturers = -0.47, 90% CI: -1.07 to 0.13) and will to compete (standardized estimates: early maturers = 0.13, 90% CI -0.25 to 0.51; average maturers = -0.09, 90% CI: -0.45 to 0.24; late maturers = -0.27, 90% CI: -0.83 to 0.23), particularly in the older age groups, i.e., under-15 and under-17 (Fig 5). Only for competitiveness, under 13 players showed lower values for competitiveness than the other age groups (Fig 6). There was no substantial variation between players' scores by gender, age group, maturity status, and the onset of deliberate practice for achievement and competitiveness motivation (S2–S4 Figs of S1 File).

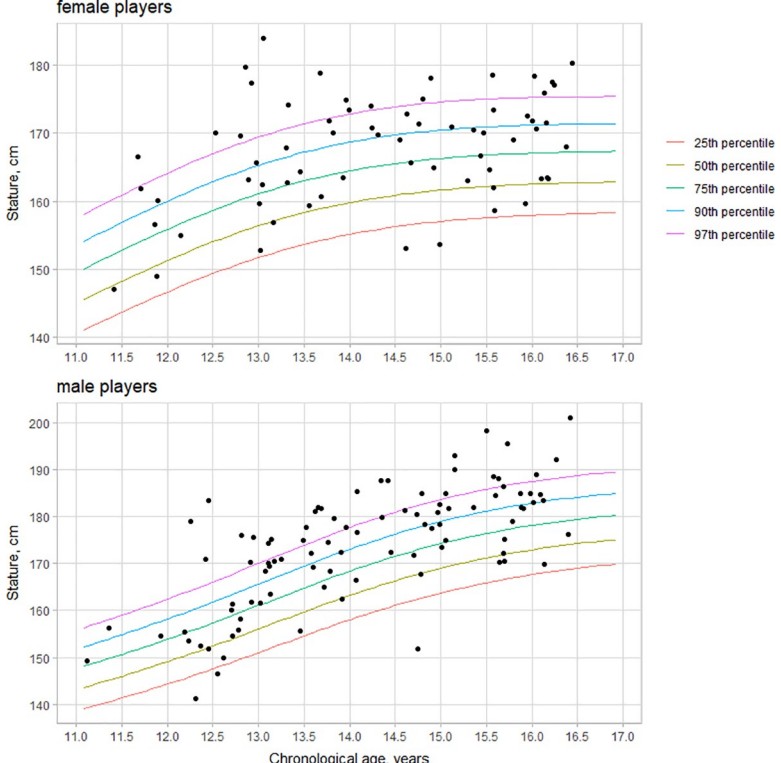

**Fig 1.** Statures of young female (upper panel) and male (lower panel) volleyball players by chronological age against the World Health Organization (WHO) growth references for stature.

**Table 2. Multilevel regression models estimates and 90% credible intervals of young Brazilian volleyball players performance and motivation adjusted by age group, maturity status, onset of deliberate practice and gender.**

| | Population-level parameter ($\beta^0$) | Group-level parameters (standard deviation) | | | | Data-level residuals ($\epsilon_i$) |
|---|---|---|---|---|---|---|
| | | Age group ($\alpha_{a[i]}^{age\ group}$) | Maturity status ($\alpha_{m[i]}^{maturity\ status}$) | onset of deliberate practice $\alpha_{d[i]}^{deliberate\ practice}$ | Gender ($\alpha_{g[i]}^{gender}$) | |
| *Performance* | | | | | | |
| Countermovement jump, cm | -0.28 (-1.79 to 1.21) | 1.00 (0.47 to 1.82) | 0.25 (0.01 to 0.81) | 0.39 (0.04 to 1.07) | 0.69 (0.17 to 1.65) | 0.78 (0.71 to 0.85) |
| 2-kg medicine ball throw, m | -0.22 (-1.74 to 1.29) | 1.04 (0.52 to 1.85) | 0.49 (0.04 to 1.33) | 0.25 (0.01 to 0.85) | 0.72 (0.19 to1.64) | 0.72 (0.65 to 0.79) |
| Sprint 10-m, s | 0.09 (-1.06 to 1.21) | 0.49 (0.10 to 1.21) | 0.24 (0.01 to 0.81) | 0.36 (0.03 to 1.06) | 0.54 (0.05 to 1.46) | 0.98 (0.89 to 1.08) |
| *Deliberate practice motivation* | | | | | | |
| Will to excel, 1–5 | -0.15 (-1.27 to 0.93) | 0.38 (0.04 to 1.06) | 0.55 (0.09 to 1.31) | 0.27 (0.01 to 0.90) | 0.40 (0.02 to1.31) | 0.97 (0.88 to 1.07) |
| Will to compete, 1–5 | -0.04 (-0.81 to 0.70) | 0.23 (0.01 to 0.74) | 0.43 (0.05 to 1.12) | 0.26 (0.01 to 0.81) | 0.39 (0.02 to 1.24) | 1.00 (0.91 to 1.11) |
| *Achievement and competitiveness motivation* | | | | | | |
| Mastery, 1–5 | -0.04 (-0.97 to 0.94) | 0.24 (0.01 to 0.81) | 0.25 (0.01 to 0.80) | 0.32 (0.02 to 0.98) | 0.42 (0.02 to 1.35) | 1.01 (0.92 to 1.11) |
| Work, 1–5 | -0.01–0.68 to 0.67) | 0.24 (0.01 to 0.76) | 0.25 (0.01 to 0.79) | 0.28 (0.01 to 0.85) | 0.37 (0.02 to 1.20) | 1.01 (0.93 to 1.11) |
| Competitiveness, 1–5 | -0.09 (-1.20 to 1.00) | 0.52 (0.11 to 1.28) | 0.31 (0.02 to 0.93) | 0.38 (0.04 to 1.05) | 0.48 (0.03 to 1.36) | 0.97 (0.89 to 1.07) |

Note: All outcomes were standardized (z-score).

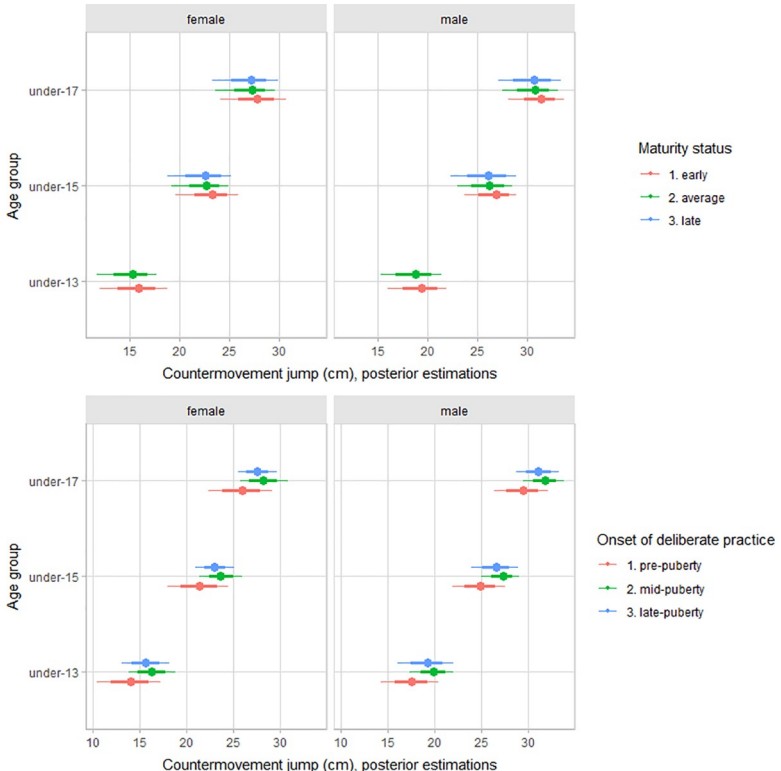

**Fig 2.** Posterior estimations and uncertainty (bold lines and thick ones represent 67% and 90% intervals, respectively) for countermovement jump performance by age group and contrasting age group by maturity status (upper plot), and the onset of deliberate practice (lower plot).

## Discussion

Studies examining the interacting influence of chronological age, biological maturity status, and accumulated deliberate practice in young athletes´ functional and psychological characteristics are scarce, particularly with female athletes. To our best knowledge, this study is the first to consider age-, maturity-, and deliberate practice-associated variation on growth, functional performance motivation for achievement, competitiveness, and deliberate practice of youth volleyball players. Furthermore, Brazil volleyball national teams have been consistently the reference of the highest adult level for at least the last two decades for both female and male players [27]. Hence, the study of youth volleyball programs in Brazil potentially offers a unique context to understand the expertise development in team sports.

To our best knowledge, available data with youth volleyball characteristics is limited. There was substantial variation in body dimensions and functional performance between young female and male Brazilian volleyball players in the present sample. Young male players were higher, heavier, and with higher performance scores than the young female players. However, there was no variation by gender for motivation for deliberate practice and motivation for achievement and competitiveness. As should be expected, sexual dimorphism needs to be accounted for in the interpretations of young volleyball athletes' body dimensions and functional capacity [42]. However, the present sample's young volleyball players appeared to be highly motivated and committed to deliberate practice, achievement, and competitiveness. The present data suggest that the Brazilian youth volleyball training environment also seems to contribute to players being motivated and engaged with deliberate practice, independent of

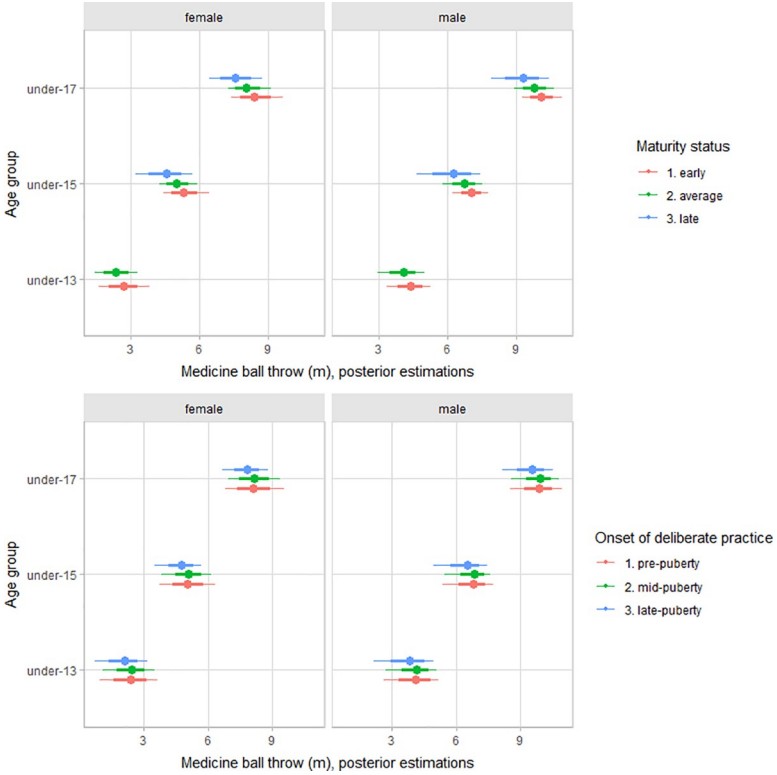

**Fig 3.** Posterior estimations and uncertainty (bold lines and thick ones represent 67% and 90% intervals, respectively) for 2-kg medicine ball throw performance by age group and contrasting age group by maturity status (upper plot), and the onset of deliberate practice (lower plot).

gender. Given the sustained excellence of Brazilian volleyball at the adult level, it may be reasonable to consider that the young Brazilian volleyball players may be oriented towards competitive success and exhibit a strong will to become expert players.

On average, most of the sample young Brazilian volleyball players were above the 75[th] percentile age- and gender-specific reference of the WHO growth references [40], which include Brazilian data [43]. In many cases, statures were above the 97[th] percentiles for female and male players (S1 Fig of S1 File). The mean masses of male and female youth Brazilian volleyball players with age and gender-specific were between 50[th]-90[th] percentiles of the US population [41]. There is no comparable data available for body mass in the WHO growth references [40]. Available data with young volleyball players is scarce. The young Brazilian volleyball players' body size was similar to Australian adolescent volleyball players from different competitive levels, i.e., national, state, and novice [12], and the England men's junior volleyball teams [44]. Body dimensions appear to be highly valued in the selection process of youth volleyball, in particular stature. Interpretation of body dimensions during the pubertal growth period may be problematic, as the variation between individuals is considerable. The transient size advantages of early maturing players may be overvalued with a naïve interpretation.

We used the gender-specific maturity offset equations in the present study, which were recently simplified [28]. The offset equations estimate individuals´ distance to PHV, providing an estimate of their maturity status. The equations give an alternative to having a reference of maturity status when considering cross-sectional observations. However, the equations, the original [45] and simplified version [28], have limited validity [30]. Hence, the protocol may not be a sufficiently sensitive indicator of maturity status requiring a conservative

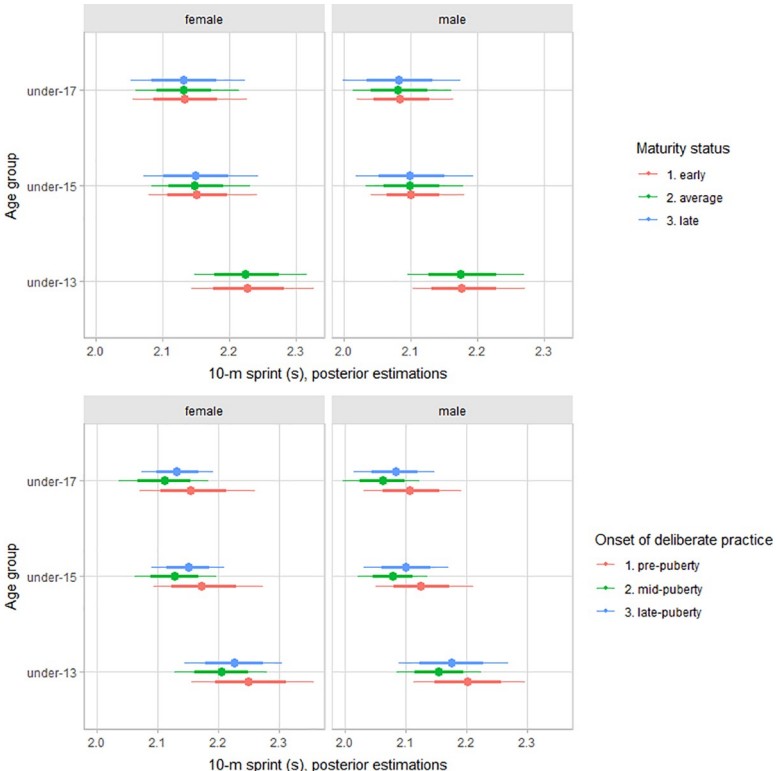

**Fig 4.** Posterior estimations and uncertainty (bold lines and thick ones represent 67% and 90% intervals, respectively) for 10-m sprint performance by age group and contrasting age group by maturity status (upper plot), and the onset of deliberate practice (lower plot).

interpretation of the data. In our sample, both female and male volleyball players appeared to be mostly early or average maturers. The present data suggest that early to average maturing and taller girls and boys may be advantaged to be retained within youth volleyball programs. The results may reflect selection or exclusion (self, coach, or some combination), the different success of players advanced in maturation, the changing nature of the game (the increase of the net height in older age groups), or some combination of these factors [46]. A similar trend has been noted in youth sports where body dimensions are determinants of performance [36].

Interestingly, the late-maturing players in our sample, regardless of gender, had a late-onset of deliberate volleyball practice. Furthermore, both female and male players' functional performance and motivation scores, except competitiveness, did not vary substantially by maturity status, adjusting for age group, and the onset of deliberate practice. Given that late-maturing individuals may have a more significant potential to attain higher adult stature [47], youth volleyball coaches should consider players' maturity status to help their interpretations about players' physique, performance, and behavior.

Youth sports programs often are focused on talent development and expertise attainment [48]. Often, coaches and researchers assume the need for early-onset deliberate practice during childhood and extensive accumulation of hours of training through the sports career as imperative to develop expertise in adulthood [49, 50]. Hence, early specialization in many youth sports contexts is the mainstream path for talent development and achievement of professional status in adult sports [14, 15, 51]. However, data in youth soccer [52] and volleyball [53] showed children and adolescents engaged in multi-sports developmental programs and/or with a later onset of deliberate practice attained expertise at adult levels. In volleyball, our data

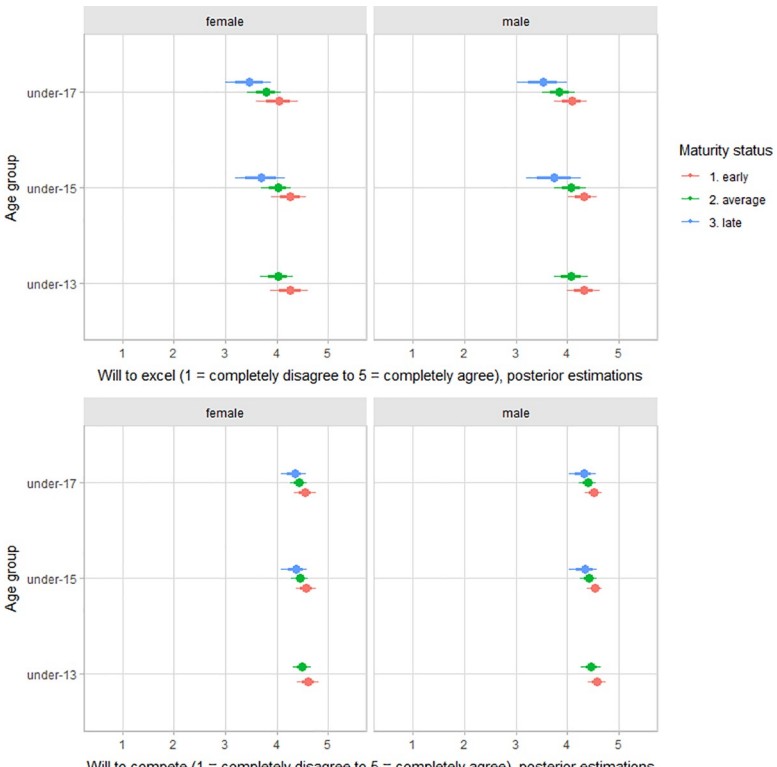

**Fig 5.** Posterior estimations and uncertainty (bold lines and thick ones represent 67% and 90% intervals, respectively) for will to excel (upper plot) and will to compete (lower plot) scores by age group and contrasting age group by maturity status.

concur with observations of the onset of deliberate practice during the pubertal years or even during late pubertal growth [53–55]. Also, the development of expertise in volleyball associated with a later onset of deliberate volleyball practice may benefit from previous multi-sports participation during childhood and early adolescence [53, 54], as postulated in the Developmental Model of Sport Participation framework [56]. However, we did not retain information about players´ previous sports participation and experiences, limiting our interpretation.

Adjusting for age group and maturity status, early accumulation of deliberate volleyball practice does have a substantial contribution to explain variation between players functional capacities, particularly for vertical jump and sprint performance. Given the importance of jump in volleyball, coaches and trainer might need to be cautious interpreting the jump performance at early ages has early advantages, interpreted as a "potential gifted athlete", probably reflecting differences in accumulated training stimulus. Our results were consistent with observations in young basketball players [20], noting that players with exposure to sport-specific deliberate practice during the pubertal years or late pubertal growth have better overall physiological performance than players with an onset of deliberate practice during childhood. Overall, our data add to the argument that early specialization in sport has not been shown to enhance physiological systems more than diversified participation in physical activity and sport [57]. On the other hand, there was no apparent relation between the early accumulation of deliberate volleyball practice with deliberate practice motivation and achievement and competitiveness motivation. Hence, our data is inconsistent with the claims that early exposure to deliberate single-sport practice decreases motivation for participation [17–19, 58], at least in youth volleyball.

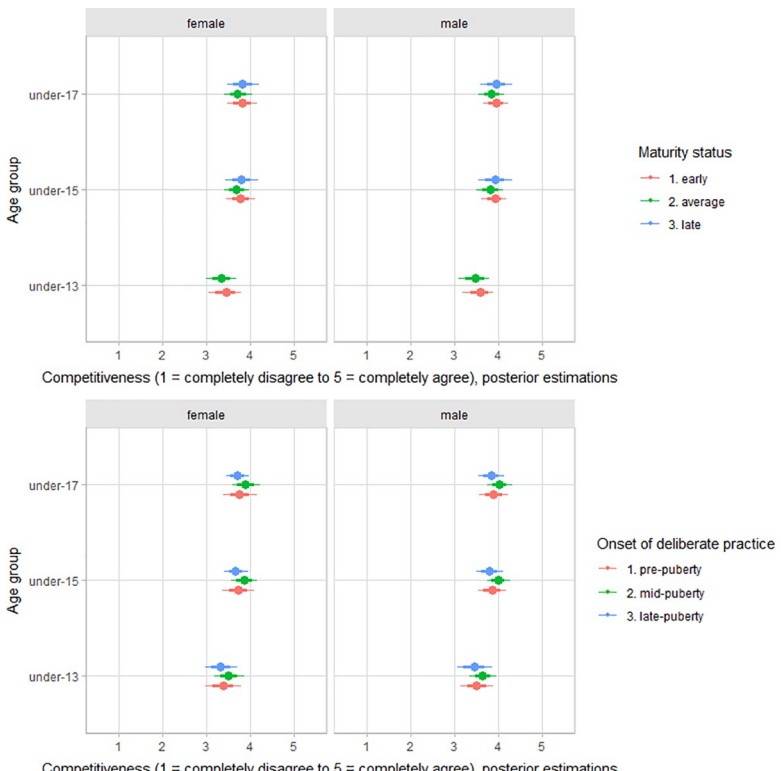

**Fig 6.** Posterior estimations and uncertainty (bold lines and thick ones represent 67% and 90% intervals, respectively) for competitiveness score by age group and contrasting age group by maturity status (upper plot), and the onset of deliberate practice (lower plot).

Inferences about young athletes´ development based on chronological age, maturity status, or accumulated training experience per se are incomplete, at best. Nevertheless, it remains a common practice of research reporting in youth sports studies. To provide more accurate interpretations, coaches and researchers should align the player's chronological with its growth pattern and with her or his accumulated sports experience. Hence, researchers should consider modeling approaches that can deal with the different levels and sources of variations (i.e., hierarchical or cross-classified structure), often based on imbalanced samples and noisy measurements. Traditional analytical approaches (e.g., t-tests, least-squares linear regression, analysis of (co)variance, and many others), i.e., single-level fixed effects regressions, are an unsatisfactory default for analysis [22]. Single-level regressions treat the units of analysis as independent observations, and in particular higher-level predictor variables will be the most affected by ignoring grouping [59]. In the alternative, multilevel modeling should be considered as a default approach, as already in several scientific areas [26]. Multilevel models allow and explicitly model the data structure by allowing for residual components at each level in the hierarchy or cluster [59], i.e., the model explicitly variations within and between units (individuals and/ or groups). Multilevel models partially pool the information across units to produce better estimates for all units in the data [26]. Nevertheless, multilevel modeling requires more attention to be used properly.

Given the need for transparency and reproducibility in science, we provide the datasets, model codes, and supplementary material supporting this study inference in an open repository (osf.io/ud2ev). Although we discuss and speculate our observations given the currently available knowledge in the literature, we acknowledge that this study represents a single

observational study, and interpretations and generalizations need to be conservative. A key feature of Bayesian inference lies in the explicit updating of knowledge based on data accumulated from several observational studies [26], particularly in scientific areas using different sources and data levels such as sport sciences. Hence, this study's data and its interpretations should be integrated with future studies to provide a more comprehensive understanding of young volleyball players' development.

## Conclusion

Conditional on the data, young Brazilian volleyball players tend to have an onset of deliberate practice during pubertal growth years or late adolescence. Researchers and coaches should consider sexual dimorphism to interpret young volleyball players' body dimensions and functional capacity. However, motivation characteristics related to deliberate practice, achievement, and competitiveness appear similar in female and male young players. The alignment of chronological age, biological maturation, and accumulated training experience in the sport may allow more in-depth insights into young volleyball players' development, providing sounder support for coaches´ decisions in youth volleyball. Hence, coaches and others involved with youth volleyball programs need to be familiar with the growth and maturation basic principles.

## Supporting information

**S1 File.**
(PDF)

## Author Contributions

**Conceptualization:** Felipe G. Mendes, Carine Collet, Carlos E. Gonçalves, Humberto M. Carvalho.

**Data curation:** Felipe G. Mendes.

**Formal analysis:** Ricardo T. Quinaud, Carlos E. Gonçalves, Humberto M. Carvalho.

**Investigation:** Felipe G. Mendes, Ahlan B. Lima, Marina Christofoletti, Ricardo T. Quinaud, Carine Collet, Carlos E. Gonçalves.

**Methodology:** Felipe G. Mendes, Ahlan B. Lima, Marina Christofoletti, Ricardo T. Quinaud, Carine Collet.

**Supervision:** Humberto M. Carvalho.

**Writing – original draft:** Felipe G. Mendes, Humberto M. Carvalho.

**Writing – review & editing:** Ahlan B. Lima, Marina Christofoletti, Ricardo T. Quinaud, Carine Collet, Carlos E. Gonçalves, Humberto M. Carvalho.

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
