## [Decision Letter · Decision Letter 0]

4 Jan 2021

PONE-D-20-27002

Multidimensional characteristics of young Brazilian volleyball players: a Bayesian multilevel analysis

PLOS ONE

Dear Dr. Carvalho,

Thank you for submitting your manuscript to PLOS ONE. After careful consideration, we feel that it has merit but does not fully meet PLOS ONE’s publication criteria as it currently stands. Therefore, we invite you to submit a revised version of the manuscript that addresses the points raised during the review process.

We look forward to receiving your revised manuscript.

Kind regards,

Nili Steinberg

Academic Editor

PLOS ONE

Journal Requirements:

2) Please include your tables as part of your main manuscript and remove the individual files. Please note that supplementary tables (should remain/ be uploaded) as separate "supporting information" files

3) Please upload a copy of Supporting Information Figure 1 which you refer to in your text on line 300.

Reviewers' comments:

Reviewer's Responses to Questions

**Comments to the Author**

1. Is the manuscript technically sound, and do the data support the conclusions?

Reviewer #1: Partly

Reviewer #2: Yes

2. Has the statistical analysis been performed appropriately and rigorously? 

Reviewer #1: No

Reviewer #2: Yes

3. Have the authors made all data underlying the findings in their manuscript fully available?

Reviewer #1: Yes

Reviewer #2: Yes

4. Is the manuscript presented in an intelligible fashion and written in standard English?

Reviewer #1: No

Reviewer #2: Yes

5. Review Comments to the Author

Reviewer #1: l118: The abbreviated term, PHV, was observed at first time here, but you did not explain it beforehand. At line 135, I found that the “peak height velocity” was abbreviated to the PHV.

l123,-123, l138-l140, l146-147:

Same information of PHV is duplicated and redundant. Please consider if it should be omitted.

l201:

How did you categorize age groups? In the equation, participants’ age seemed to be divided into four groups.

However, I could not find the descriptions of divided points for age in the Method section.

In figures, age groups were shown as under-17, under-15, and under-13. However, these were three age groups, not four.

l201, l202:

These symbols were mistakes?

for i = 1, 2, 3, 4 -> for a = 1, 2, 3, 4

for s = 1, 2,3 -> for d = 1, 2, 3

l200-l214:

I could not understand well the reason why you applied a multilevel model for this data in which each group included only several categories.

For example, when you inferred the effect of gender on outcomes, it would be enough to estimate the fixed effect of the gender (i.e., slope of the gender on the outcome) using an ordinary regression model. At least, for binary variable (i.e., gender), it seemed to be unnecessary to assume the hierarchical structure of parameters.

In addition, you categorized age into some groups, but the effect of age on outcomes should be inferred using continuous predictive variable. If you pooled continuous age values into some categories, you would lack much information about influence of age on outcome.

Taken together, it seemed that the ordinary regression model would be enough for your analysis. If not so, please explain clearly the purpose of using multilevel model for this study.

l207:

What do “population-level parameter” and “group-level parameters” mean? Is “population-level parameter” equivalent to beta? Is "group-level parameter" equivalent to alpha? If you called sigma "group-level parameter", it would be wrong. Sigma is often called “hyper parameter”. Alpha is often called “random intercept” or “varying intercept”.

What did you set a prior for each sigma?

In addition, in the result section, you should report the results of inferred group-level intercept (i.e., beta) and hyper parameters (i.e., sigma).

Figures, Table, and Figure captions:

Figures and Table do not include enough information. You should also refine them.

In figure 1, what does each color of lines mean?

How did you produce the predicted lines in Figure 1? They seemed to be produced from a nonlinear regression model, but the details of the model were not found. Please explain what “the growth references of the WHO” indicates.

In Figure 2-Figure 6, lines overlapped, and we could not detect each line. Each line should be displayed by shifting each position. In addition, I could not distinguish the 90% and 67% intervals in these Figures.

What does "mat_cat" mean?

l249-l253: I could not figure out which part we should see in each figure in order to understand “there was substantial variation between female and male players”.

Figure 6 is identical with Figure 5?

In Table 1, I could not understand what “1-5”, which was described in “Deliberate practice motivation” and “Achievement and competitiveness motivation”, meant.

Reviewer #2: The authors provide a description of the variation in body size, functional capacities, motivation for achievement and competitiveness, and motivation for deliberate practice of 68F+94M Brazilian adolescent volleyball players. The paper essentially consists of a descriptive statistics (with a Bayesian multilevel modelling for estimates' adjustment), oriented toward the assessment of different levels of motivations. The manuscript is reasonably written, and quite easy to follow also by an audience of non-specialists; from a data science perspective the paper is quite simple and straightforward, with very basic statistical approaches.

Hereafter I list a few points that the authors need to take into account and discuss further:

- aim of the research needs to be better clarified - the descriptive purposes is rather vague and poorly focused

- the use of the Bayesian modelling should be explained in more details, both in terms of motivation and in its mathematical application in the current situation

- the sample size is too small to allow for robust generalisation of the results - in the discussion the conclusions drawn are beyond the support provided by the shown results.

- the term "credible interval" is somehow non-standard; what's the mathematical definition of "credible"? I would suggest using the word "confidence" instead, specifying the statistics used (e.g. 95% bootstrap t-Student)

- the usefulness of the results for volleyball trainers and specialists should be better pointed out, for instance providing some case studies highlighting how decision can be formulated by using the obtained outcomes.

- the use of amateur level player can represent a limiting factor of the study; in the authors' opinion, what are the differences to be expected if pro-level athletes are used instead for the same study?

6. PLOS authors have the option to publish the peer review history of their article (what does this mean?). If published, this will include your full peer review and any attached files.

Reviewer #1: No

Reviewer #2: No

---

## [Author Response · Author response to Decision Letter 0]

16 Feb 2021

Dear Prof. Nili Steinberg 

We addressed your requests to ensure we followed PLOS ONE's style requirements, including those for file naming, include the table as part of your main manuscript and uploaded as separate "supporting information" the supporting figures. We specifically upload S1 fig wich is referenced in the text.

Hope we appropriatly reply to your requests.

Best regards

Humberto Carvalho

---

## [Decision Letter · Decision Letter 1]

2 Mar 2021

PONE-D-20-27002R1

Multidimensional characteristics of young Brazilian volleyball players: a Bayesian multilevel analysis

PLOS ONE

Dear Dr. Carvalho,

Thank you for submitting your manuscript to PLOS ONE. After careful consideration, we feel that it has merit but does not fully meet PLOS ONE’s publication criteria as it currently stands. Therefore, we invite you to submit a revised version of the manuscript that addresses the points raised during the review process.

We look forward to receiving your revised manuscript.

Kind regards,

Nili Steinberg

Academic Editor

PLOS ONE

Reviewers' comments:

Reviewer's Responses to Questions

**Comments to the Author**

1. If the authors have adequately addressed your comments raised in a previous round of review and you feel that this manuscript is now acceptable for publication, you may indicate that here to bypass the “Comments to the Author” section, enter your conflict of interest statement in the “Confidential to Editor” section, and submit your "Accept" recommendation.

Reviewer #1: (No Response)

Reviewer #2: All comments have been addressed

2. Is the manuscript technically sound, and do the data support the conclusions?

Reviewer #1: Partly

Reviewer #2: Yes

3. Has the statistical analysis been performed appropriately and rigorously? 

Reviewer #1: No

Reviewer #2: Yes

4. Have the authors made all data underlying the findings in their manuscript fully available?

Reviewer #1: Yes

Reviewer #2: Yes

5. Is the manuscript presented in an intelligible fashion and written in standard English?

Reviewer #1: Yes

Reviewer #2: Yes

6. Review Comments to the Author

Reviewer #1: Thank you for your response and revising the manuscript. Some of the concerns I raised earlier have been solved. However, some points which I could not understand well left. Please consider the followings.

>Authors ´ reply: We appreciate the reviewer's comment. Ordinary single-level regression would be innapropriate to deal with the data because:

>(i) the context of observation that presents a cross-classified nesting, as players belong to multiple groups (eg, a late maturing boy from the under 13 age group with an onset of deliberate practice during the pubertal years). Often analysis of covariance is used to interpret the youth sports data, likely overfitting the data, as multiple comparisons are highly problematic using a single-level model, even more in settings with a low group-level variation (Gelman, Hill, & Yajima, 2012);

>(ii) we examined the need to consider multilevel models by looking into null models (the simplest two level model which includes only the group-level parameters (also referred to as random parameters), to measure the proportion of total variance which fell between-groups (i.e., variance partition coefficient also reffered to as intracalss correlation)(Goldstein, 2011). Variance partition correlations higher than 0.05 indicate a substantial variation by the respective group and the need to use multilevel modeling (Goldstein, 2011; Snijders and Bosker, 2012).

>(iii) the key feature of multilevel modelling is shrinkage (partial pooling), by incorporating group- level effects. It allows for more efficient use of the data. These may be understood as a weighted average between the all sample estimate and a group or unit estimate. The specific weights are based on the entire variation of the sample and the group variation (Gelman and Hill, 2007). In particular, the partial pooling will be stronger for smaller units with fewer observations (Buttice and Highton, 2013)

>We agree with the argument to interpret the age effect assuming chronological age in the model. Indeed categorization removes potential variation. We choose to use age groups to echo the ecological context where players are grouped in competitive age groups. Also, following our previous point, partial pooling helps to mitigate the limitations of our choice for group age.

I agree all of advantages of Bayesian analysis you have raised. In addition, I also agree justification of categorizing age into several groups.

However, it is still odd for me that you have assumed the hierarchical structure of effect of gender. I cannot understand well the sense of estimating the variance of the effect between gender, which includes only two groups. Estimating "variance" of normal distribution, which the group including only two categories obeys, seems to be equivalent to estimating the "difference" between the two categories.

For the group including only two categories, in contrast to the regression model using the group as a dummy, the multilevel model seems to contain redundant parameters (i.e., each of individual parameters and standard deviation of normal distribution). The predictive accuracy might not be improved drastically, even if the multilevel model was used. It would not be necessary to consider partial pooling (shrinkage) for gender logically because there are only two groups and unknown groups do not exist.

In order to demonstrate whether there is a variance between male and female (in other words, whether there is a difference between the two), estimating the fixed slope of the gender seems to be enough.

If you do not agree, in your response, please refer to previous works which assumed the multilevel model for groups which include only two groups.

>Authors ´ reply: We used referred to the hyperparameters as population-level effects, which are referred to as fixed effects in frequentist analysis, and group-level effects which are referred to as random effects in frequentist analysis. This is the common designation when using Bayesian analysis. We inserted the comparable frequentist terms in the text to allow the unfamiliar reader with Bayesian analysis to better understand our report. For the residual standard deviation parameter (level-1 standard deviation or referred to as residuals), we adopted the default prior in the brms package. For the group-level effects (i.e., level-2 standard deviations, in this case,,,) we used half-normal priors, as we intend to have a conservative approach on the analysis by stating that the group-level estimates are unlikely to be greater than one standard deviation of the outcome.

I understood that the population-level effects and the group-effects represented the fixed effects (i.e., the intercept of your regression model) and random effects (i.e., sigma_a, sigma_m, sigma_d, and sigma_g), respectively.

You have not report the summary of inferred group-level intercept and group-effects yet, although I suggested earlier. Summary of posterior distributions of estimated parameters (e.g., mean, standard deviation, and 95% CI) may be necessary. As I suggested in the following comment, not only visualization of the results but also numeric information would be needed for objective explanations.

Is the "residual standard deviation parameter (level-1 standard deviation)" equivalent to the standard deviation of the normal distribution which the response variable (i.e., y_{i}) obeys?

In the model, information of the distributions which the response variable obeys is lack. The "level-1 standard deviation" should be shown clearly in the equation of the model.

For instance,

\\hat{y}_{i} = \\beta + \\alpha_{a[i]} + ...

...

y_{i} \\sim Normal(hat_{y}_{i}, \\sigma)

In addition, please explain the "default prior" in brms package concretely (i.e., the name of probability distribution and parameters).

In your response, half-normal priors were set for the group-level effects. However, in the revised manuscript, you described that normal prior (0, 5) was set as prior for group-level parameters. You should clearly describe "half-normal".

>Authors ´reply: We thank the reviewer ´s comment. Hopefully, we provide cleaner and clear Figures now. We dodged the estimates and uncertainty to remove the overlap. The legend titles were corrected in all figures, removing the variable label from the dataset.

For captions in Figure 2-6, you should explain that the bold lines and thick ones represent 67% and 90% intervals, respectively.

The legend of the deliberate practice disappeared in Figure 5, whereas it appeared in other figures. I found that the relationship between the onset of deliberate practice and will to excel (or will to compete) was reported in Figure S2. However, it seemed that S2-S4 Figs were never referred to in the main text.

>Authors ´reply: We apologize for the incorrect upload of the figures, We have corrected appropriately both Figure 5 and Figure 6. The Bayesian inference allows for direct probabilistic comparisons between estimates and uncertainty. Hopefully, with the changes in the Figures, it becomes clear for the reader.

I could not still understand well which of the parts in each figure indicated “substantial variation between female and male players”. Intervals shown in each panel seemed to indicate "variation between players within each gender". The results for each gender were separately shown by divided panels: therefore, it was difficult to compare differences between two genders.

In addition, because the interpretation using only visualized results seemed to be subjective and arbitrary, quantitative reports such as numeric information of intervals of parameters may be necessary. The posterior distributions of each parameter (e.g., group-level parameters such as sigma_gender, or slope of the gender in the regression model) should be reported for the objective interpretations.

>Authors ´reply: It represents the scale used in the questionnaires based on a Likert-like scale. The information about the questionnaires is provided in the methods section. It is a common practice to describe questionnaire scales, but we are open to suggestions to improve our reporting.

Please show the meaning of the number: for example, "(1: disagree, 5: agree)". Similarly, in the method and materials section, it should be described the labels of each number. We could not understand what the greater number meant, if the descriptions were omitted.

Reviewer #2: (No Response)

7. PLOS authors have the option to publish the peer review history of their article (what does this mean?). If published, this will include your full peer review and any attached files.

Reviewer #1: No

Reviewer #2: No

---

## [Author Response · Author response to Decision Letter 1]

23 Mar 2021

Specific reply to reviewer´s comments 

Authors´ reply: We appreciate the reviewer´s comments and suggestions. It has been uncommon to have comments to our models that allow us to rethink or correct them and the interpretations. Hopefully, we satisfactorily addressed the reviewer´s concerns.

Reviewer #1: Thank you for your response and revising the manuscript. Some of the concerns I raised earlier have been solved. However, some points which I could not understand well left. Please consider the followings.

>Authors ´ reply: We appreciate the reviewer's comment. Ordinary single-level regression would be inappropriate to deal with the data because:

>(i) the context of observation that presents a cross-classified nesting, as players belong to multiple groups (eg, a late maturing boy from the under 13 age group with an onset of deliberate practice during the pubertal years). Often analysis of covariance is used to interpret the youth sports data, likely overfitting the data, as multiple comparisons are highly problematic using a single-level model, even more in settings with a low group-level variation (Gelman, Hill, & Yajima, 2012);

>(ii) we examined the need to consider multilevel models by looking into null models (the simplest two level model which includes only the group-level parameters (also referred to as random parameters), to measure the proportion of total variance which fell between-groups (i.e., variance partition coefficient also reffered to as intracalss correlation)(Goldstein, 2011). Variance partition correlations higher than 0.05 indicate a substantial variation by the respective group and the need to use multilevel modeling (Goldstein, 2011; Snijders and Bosker, 2012).

>(iii) the key feature of multilevel modelling is shrinkage (partial pooling), by incorporating group- level effects. It allows for more efficient use of the data. These may be understood as a weighted average between the all sample estimate and a group or unit estimate. The specific weights are based on the entire variation of the sample and the group variation (Gelman and Hill, 2007). In particular, the partial pooling will be stronger for smaller units with fewer observations (Buttice and Highton, 2013)

>We agree with the argument to interpret the age effect assuming chronological age in the model. Indeed categorization removes potential variation. We choose to use age groups to echo the ecological context where players are grouped in competitive age groups. Also, following our previous point, partial pooling helps to mitigate the limitations of our choice for group age.

I agree all of advantages of Bayesian analysis you have raised. In addition, I also agree justification of categorizing age into several groups.

However, it is still odd for me that you have assumed the hierarchical structure of effect of gender. I cannot understand well the sense of estimating the variance of the effect between gender, which includes only two groups. Estimating "variance" of normal distribution, which the group including only two categories obeys, seems to be equivalent to estimating the "difference" between the two categories.

For the group including only two categories, in contrast to the regression model using the group as a dummy, the multilevel model seems to contain redundant parameters (i.e., each of individual parameters and standard deviation of normal distribution). The predictive accuracy might not be improved drastically, even if the multilevel model was used. It would not be necessary to consider partial pooling (shrinkage) for gender logically because there are only two groups and unknown groups do not exist.

In order to demonstrate whether there is a variance between male and female (in other words, whether there is a difference between the two), estimating the fixed slope of the gender seems to be enough.

If you do not agree, in your response, please refer to previous works which assumed the multilevel model for groups which include only two groups.

Authors´ reply: We thank the reviewer´s comment and understand the reviewer´s point. There is no substantial gain in the predictive accuracy with our data by include gender and population- or group-level parameters. The first reason we retained varying intercepts by gender was that it makes more sense from a biological perspective to allow for scores to vary by gender, assuming a 2 ×3 ×3 ×3 structure as 54 exchangeable groups. The multilevel approach allows us to estimate the parameters as varying effects, taking advantage of the multilevel structure (cross-classified) of the data. The general approach for multilevel models gives each batch of regression coefﬁcients with greater than two groups an independent normal distribution centered at 0 and with standard deviation estimated from data [1]. Again we recognize that there is limited gain to multilevel modeling for batches with smaller than three groups when prior distributions are noninformative [2], and are often used as population level for computational convenience.

Nevertheless, the inclusion of the variables with two batches as group-level parameters is common, particularly with the increase of computation efficiency [3-7]. However, we do not use noninformative priors, as given the standardization of the outcomes, the priors used for the group-level parameters we expect that the group-level estimates are unlikely to be greater than one standard deviation of the outcome. This choice likely adds a more conservative estimation, while using all available information in the data (and at least to as reflecting better the research context). We hope to have address the reviewer´s comments.

>Authors ´ reply: We used referred to the hyperparameters as population-level effects, which are referred to as fixed effects in frequentist analysis, and group-level effects which are referred to as random effects in frequentist analysis. This is the common designation when using Bayesian analysis. We inserted the comparable frequentist terms in the text to allow the unfamiliar reader with Bayesian analysis to better understand our report. For the residual standard deviation parameter (level-1 standard deviation or referred to as residuals), we adopted the default prior in the brms package. For the group-level effects (i.e., level-2 standard deviations, in this case,,,) we used half-normal priors, as we intend to have a conservative approach on the analysis by stating that the group-level estimates are unlikely to be greater than one standard deviation of the outcome.

I understood that the population-level effects and the group-effects represented the fixed effects (i.e., the intercept of your regression model) and random effects (i.e., sigma_a, sigma_m, sigma_d, and sigma_g), respectively.

You have not report the summary of inferred group-level intercept and group-effects yet, although I suggested earlier. Summary of posterior distributions of estimated parameters (e.g., mean, standard deviation, and 95% CI) may be necessary. As I suggested in the following comment, not only visualization of the results but also numeric information would be needed for objective explanations.

Is the "residual standard deviation parameter (level-1 standard deviation)" equivalent to the standard deviation of the normal distribution which the response variable (i.e., y_{i}) obeys?

In the model, information of the distributions which the response variable obeys is lack. The "level-1 standard deviation" should be shown clearly in the equation of the model.

For instance,

\\hat{y}_{i} = \\beta + \\alpha_{a[i]} + ...

...

y_{i} \\sim Normal(hat_{y}_{i}, \\sigma)

In addition, please explain the "default prior" in brms package concretely (i.e., the name of probability distribution and parameters).

Authors´ reply: 

We agree with the reviewer´s point. The multilevel equation was incomplete without the level-1 residual parameter. We added the level 1 residual. We also added in the text the “brms” default prior used, Student-t (3, 0, 2.5). We added a summary of the posterior distributions as a table, as suggested. 

In your response, half-normal priors were set for the group-level effects. However, in the revised manuscript, you described that normal prior (0, 5) was set as prior for group-level parameters. You should clearly describe "half-normal".

Authors´ reply: We apologize if we were incorrect in our response. We used a normal prior (0,1) for the group parameters. It was correctly stated in the manuscript. By default “brms” sets their left boundary to zero, to keep the HMC algorithm from exploring negative variance values. But formally, we state the definition of a normal prior in the equation and in “brms” code. Hopefully, our reply is now clear. 

>Authors ´reply: We thank the reviewer ´s comment. Hopefully, we provide cleaner and clear Figures now. We dodged the estimates and uncertainty to remove the overlap. The legend titles were corrected in all figures, removing the variable label from the dataset.

For captions in Figure 2-6, you should explain that the bold lines and thick ones represent 67% and 90% intervals, respectively.

The legend of the deliberate practice disappeared in Figure 5, whereas it appeared in other figures. I found that the relationship between the onset of deliberate practice and will to excel (or will to compete) was reported in Figure S2. However, it seemed that S2-S4 Figs were never referred to in the main text. 

Authors´ reply: We added the information about the uncertainty intervals in the figure caption. In figure 5, contrasts are presented by age group and maturity status. Hence we represented only the legend for maturity status, and give the information in the figure caption. We also added S5 Fig that was missing. We added information in the results section to refer to the supplementary material. 

>Authors ´reply: We apologize for the incorrect upload of the figures, We have corrected appropriately both Figure 5 and Figure 6. The Bayesian inference allows for direct probabilistic comparisons between estimates and uncertainty. Hopefully, with the changes in the Figures, it becomes clear for the reader.

I could not still understand well which of the parts in each figure indicated “substantial variation between female and male players”. Intervals shown in each panel seemed to indicate "variation between players within each gender". The results for each gender were separately shown by divided panels: therefore, it was difficult to compare differences between two genders.

In addition, because the interpretation using only visualized results seemed to be subjective and arbitrary, quantitative reports such as numeric information of intervals of parameters may be necessary. The posterior distributions of each parameter (e.g., group-level parameters such as sigma_gender, or slope of the gender in the regression model) should be reported for the objective interpretations.

Authors´ reply: As suggested, we added estimates in the text to allow for direct comparisons, also with the models estimates in Table 2. We also were conservative in the qualification of variation between groups, and interpret mainly comparing the 90% intervals. As we report the figures in the questionnaire scales for the questionnaires, we reported in the text the standardized estimate to allow a complete interpretation of the results to the reader. 

>Authors ´reply: It represents the scale used in the questionnaires based on a Likert-like scale. The information about the questionnaires is provided in the methods section. It is a common practice to describe questionnaire scales, but we are open to suggestions to improve our reporting.

Please show the meaning of the number: for example, "(1: disagree, 5: agree)". Similarly, in the method and materials section, it should be described the labels of each number. We could not understand what the greater number meant, if the descriptions were omitted.

Authors´ reply: We agree. We added both in the methods section and in the figures scale about the meaning of the likert-like scale (1 = completely disagree to 5 = completely agree). We also re-scaled the estimates to the original scale to allow a direct interpretation to the reader. 

References:

1. Park DK, Gelman A, Bafumi J (2004) Bayesian Multilevel Estimation with Poststratification: State-Level Estimates from National Polls. Political Analysis 12: 375-385.

2. Gelman A (2006) Prior distributions for variance parameters in hierarchical models (comment on article by Browne and Draper). 515-534.

3. Warshaw C, Rodden J (2012) How Should We Measure District-Level Public Opinion on Individual Issues? The Journal of Politics 74: 203-219.

4. Leemann L, Wasserfallen F (2020) Measuring Attitudes – Multilevel Modeling with Post-Stratification (MrP). In: Curini L, Franzese R, editors. The SAGE Handbook of Research Methods in Political Science and International Relations. 1st ed. Los Angeles, CA: SAGE. pp. 371-384.

5. Claassen C, Traunmüller R (2020) Improving and Validating Survey Estimates of Religious Demography Using Bayesian Multilevel Models and Poststratification. Sociological Methods & Research 49: 603-636.

6. Loux T, Nelson EJ, Arnold LD, Shacham E, Schootman M (2019) Using multilevel regression with poststratification to obtain regional health estimates from a Facebook-recruited sample. Annals of Epidemiology 39: 15-20.e15.

7. Kiewiet de Jonge CP, Langer G, Sinozich S (2018) Predicting State Presidential Election Results Using National Tracking Polls and Multilevel Regression with Poststratification (MRP). Public Opinion Quarterly 82: 419-446.

---

## [Decision Letter · Decision Letter 2]

19 Apr 2021

Multidimensional characteristics of young Brazilian volleyball players: a Bayesian multilevel analysis

PONE-D-20-27002R2

Dear Dr. Carvalho,

We’re pleased to inform you that your manuscript has been judged scientifically suitable for publication and will be formally accepted for publication once it meets all outstanding technical requirements.

Kind regards,

Nili Steinberg

Academic Editor

PLOS ONE

Additional Editor Comments (optional):

Reviewers' comments:

Reviewer's Responses to Questions

**Comments to the Author**

1. If the authors have adequately addressed your comments raised in a previous round of review and you feel that this manuscript is now acceptable for publication, you may indicate that here to bypass the “Comments to the Author” section, enter your conflict of interest statement in the “Confidential to Editor” section, and submit your "Accept" recommendation.

Reviewer #1: All comments have been addressed

2. Is the manuscript technically sound, and do the data support the conclusions?

Reviewer #1: Yes

3. Has the statistical analysis been performed appropriately and rigorously? 

Reviewer #1: Yes

4. Have the authors made all data underlying the findings in their manuscript fully available?

Reviewer #1: Yes

5. Is the manuscript presented in an intelligible fashion and written in standard English?

Reviewer #1: Yes

6. Review Comments to the Author

Reviewer #1: Thank you for sending the revised manuscript. All concerns I have raised are solved and readability has also been improved .

7. PLOS authors have the option to publish the peer review history of their article (what does this mean?). If published, this will include your full peer review and any attached files.

Reviewer #1: No

---

## [Editor Report · Acceptance letter]

21 Apr 2021

PONE-D-20-27002R2 

Multidimensional characteristics of young Brazilian volleyball players: a Bayesian multilevel analysis 

Dear Dr. Carvalho:

I'm pleased to inform you that your manuscript has been deemed suitable for publication in PLOS ONE. Congratulations! Your manuscript is now with our production department. 

Kind regards, 

on behalf of

Dr. Nili Steinberg 

Academic Editor

PLOS ONE